# Griffithsin carrageenan fast dissolving inserts prevent SHIV HSV-2 and HPV infections in vivo

Nina Derby[1], Manjari Lal[2], Meropi Aravantinou[1], Larisa Kizima[1], Patrick Barnable[1], Aixa Rodriguez[1], Manshun Lai[2], Asa Wesenberg[1], Shweta Ugaonkar[1], Keith Levendosky[1], Olga Mizenina[1], Kyle Kleinbeck[1], Jeffrey D. Lifson[3], M. Melissa Peet[4], Zachary Lloyd[4], Michael Benson[4], Walid Heneine[5], Barry R O'Keefe [6], Melissa Robbiani[7], Elena Martinelli[1], Brooke Grasperge[8], James Blanchard[8], Agegnehu Gettie[9], Natalia Teleshova[1], José A. Fernández-Romero[1,10] & Thomas M. Zydowsky[1]

Human immunodeficiency virus (HIV) pre-exposure prophylaxis (PrEP) strategies with proven in vivo efficacy rely on antiretroviral drugs, creating the potential for drug resistance and complicated treatment options in individuals who become infected. Moreover, on-demand products are currently missing from the PrEP development portfolio. Griffithsin (GRFT) is a non-antiretroviral HIV entry inhibitor derived from red algae with an excellent safety profile and potent activity in vitro. When combined with carrageenan (CG), GRFT has strong activity against herpes simplex virus-2 (HSV-2) and human papillomavirus (HPV) in vitro and in vivo. Here, we report that GRFT/CG in a freeze-dried fast dissolving insert (FDI) formulation for on-demand use protects rhesus macaques from a high dose vaginal SHIV SF162P3 challenge 4 h after FDI insertion. Furthermore, the GRFT/CG FDI also protects mice vaginally against HSV-2 and HPV pseudovirus. As a safe, potent, broad-spectrum, on-demand non-antiretroviral product, the GRFT/CG FDI warrants clinical development.

[1] Center for Biomedical Research, Population Council, 1230 York Ave, New York, NY 10065, USA. [2] PATH, 2201 Westlake Ave, Suite 200, Seattle, WA 98121, USA. [3] AIDS and Cancer Virus Program, Leidos Biomedical Research, Inc., Frederick National Laboratory for Cancer Research, 8560 Progress Dr, Frederick, MD 21701, USA. [4] MPI Research, 54943 N. Main St, Mattawan, MI 49071, USA. [5] Centers for Disease Control, 1600 Clifton Rd, Atlanta, GA 30333, USA. [6] Molecular Targets Program, Center for Cancer Research, and Natural Products Branch, Developmental Therapeutics Program, Division of Cancer Treatment and Diagnosis, National Cancer Institute, Building 560, Room 21-105, Frederick, MD 21702-1201, USA. [7] MJR4CONSULTING, New York 10065 NY, USA. [8] Tulane National Primate Research Center, 18703 Three Rivers Rd, Covington, LA 70433-8915, USA. [9] Aaron Diamond AIDS Research Center, 455 1st Ave. #7, New York, NY 10016, USA. [10] Science Department, Borough of Manhattan Community College, 199 Chambers St, New York, NY 10007, USA. These authors contributed equally: Nina Derby, Natalia Teleshova, José A. Fernández-Romero, Thomas M. Zydowsky. Correspondence and requests for materials should be addressed to N.D. (email: nderby@popcouncil.org)

The promise for success of oral pre-exposure prophylaxis (PrEP) in preventing HIV acquisition is threatened by the side effects and systemic accumulation of antiretroviral drugs (ARVs). Side effects are less acceptable in uninfected than HIV infected people. Drug accumulation may have consequences for HIV treatment in people who become infected and long-term health consequences for those who remain uninfected. ARV candidates also dominate the topical microbicide arena, bringing the same issues of side effects, long-term consequences, and resistance. Currently, the microbicide development pipeline contains no strictly non-ARV options[1].

Development of non-ARV microbicides initially centered on molecules with non-specific modes of action, but the candidates either were too weak to show efficacy in humans or caused epithelial damage, increasing HIV risk[2]. Lectins represent a specific non-ARV approach to HIV prevention, binding envelope glycans and interfering with the interactions between the envelope glycoproteins and cellular receptors. Although one lectin, cyanovirin, reduced vaginal SHIV infection in macaques[3,4], its unacceptable safety profile diminished enthusiasm for the approach[5,6].

Griffithsin (GRFT) is a mannose binding lectin derived from red algae that has an excellent safety profile[5–7]. It is the most potent anti-HIV lectin identified to date and among the most potent anti-HIV agents[8,9]. GRFT prevents both cell-free and cell-associated HIV transmission and virus-cell fusion[10] with picomolar activity against cell-free virus in vitro (50% effective concentration [$EC_{50}$] 1.6 ng/ml [0.13 nM], $EC_{90}$ 7.2 ng/ml [0.58 nM][11]). GRFT inhibits infection with other pathogens, including HSV by targeting entry and cell-to-cell transmission, and HPV by mediating receptor internalization[7,12]. GRFT's activity against sexually transmitted infections (STIs) that increase HIV susceptibility and exhibit intertwined epidemiology with HIV infection (such as HSV-2 and HPV[13–18]) adds to its appeal. A multipurpose prevention technology (MPT) that can simultaneously protect against multiple STIs may improve adherence[14]. Tackling HIV, HSV-2, and HPV with a single strategy may also improve anti-HIV efficacy.

Carrageenan (CG) is an algae-derived polysaccharide that is safe and highly potent against HPV[19–25] and is in clinical testing for HPV prevention and clearance[21,25]. The GRFT/CG combination acts synergistically against HSV[12]. Neither GRFT nor CG is readily absorbed after topical administration[26], making these antiviral agents ideal for repeated/extended topical use. CG is already included in many foods and personal care products[2] and is generally recognized as safe (GRAS), which simplifies the regulatory pathway for GRFT/CG products.

In this study, we demonstrate in vivo anti-HIV efficacy of GRFT using the SHIV SF162P3 infection model in rhesus macaques. For on-demand protection, we use a novel fast dissolving vaginal insert (FDI) formulation of GRFT/CG that stabilizes GRFT[27]. We also show that the formulation protects against HSV-2 and HPV infections in mice and provide toxicological and immunological data on repeated dosing that confirm the safety of GRFT/CG. We find that GRFT concentrations 3 logs above the in vitro $EC_{90}$ are associated with in vivo protection from SHIV and are sustained for at least 8 h after FDI insertion. A potent non-ARV on-demand MPT, the GRFT/CG FDI could have an important place within the HIV prevention toolbox.

## Results

### GRFT CG FDIs protect from SHIV SF162P3 vaginal infection.
GRFT/CG FDIs containing 1 mg GRFT (3.3 wt.%) and 3 mg CG (10 wt.%) (Table 1) protected 8 out of 10 macaques from SHIV SF162P3 infection in a vaginal challenge model. In contrast, control (CG only) FDIs protected 0 out of 10 macaques

(Fig. 1a–c). GRFT/CG and CG FDIs were inserted vaginally in depot medroxyprogesterone acetate (DMPA)-treated macaques 4 h before intravaginal challenge with 300 50% tissue culture infectious doses ($TCID_{50}$) of a viral stock containing $4.1 \times 10^3$ $TCID_{50}$ and $1.3 \times 10^7$ RNA genomes/ml. This protection was highly significant ($p = 0.0004$, Fisher's exact test); GRFT/CG FDI use resulted in a 5-fold reduction in the relative risk of infection (95% confidence interval [CI] 1.4-17.3). GRFT/CG FDIs did not appear to influence the course of infection in the 2 animals that became infected (Fig. 1d, e). Initial sequencing results (for 1 GRFT-exposed macaque and 1 control) suggest that there was no GRFT-related selection for transmitted variants within the SHIV stock (Supplementary Fig. 1, Supplementary Methods). SHIV challenge was performed in the absence of seminal fluids. However, in vitro, semen does not impact the anti-HIV activity of GRFT (Fig. 2).

### GRFT CG FDIs release GRFT vaginally without inflammation.
The GFRT/CG FDI tested against SHIV SF162P3 in vivo was selected based on in vitro stability and release criteria[27]. Since DMPA can influence mucosal drug absorption characteristics[28], we determined GRFT release in vivo both in DMPA-treated and non-DMPA treated macaques. FDIs inserted vaginally in non-DMPA-treated macaques ($n = 6$ per time point) delivered high concentrations of GRFT to the vaginal lumen. GRFT concentrations in cervicovaginal lavages (CVLs) were sustained between 1 and 8 h post-insertion while GRFT was not detected in plasma (Supplementary Fig. 2, Fig. 3a). In CVLs from macaques treated with DMPA 4 weeks before FDI insertion ($n = 6$ per time point), mean GRFT levels were also high–approximately 7000 and 4000 times the $EC_{90}$ at 4 and 8 h, respectively. Although GRFT levels at each time point were significantly higher in non-DMPA-treated than DMPA-treated macaques, the concentrations in DMPA-treated animals protected against SHIV challenge in absence of any systemic GRFT detected.

The in-vitro anti-HIV activity of CVLs from non-DMPA-treated (Fig. 3b) and DMPA-treated (Fig. 3c) FDI-treated macaques correlated tightly with the GRFT concentrations therein. The CVLs from 4 h post-insertion also significantly inhibited SHIV SF162P3 infection in polarized human cervical explants (Fig. 3d).

GRFT/CG FDIs did not alter the macaques' vaginal pH; though as expected[29], DMPA by itself increased pH (Supplementary Fig. 3). No cytokines or chemokines were increased in CVLs from DMPA-treated macaques after GRFT/CG FDI use (Table 2). Only CCL2 levels were significantly elevated in CVLs 1 h after GRFT/CG FDI insertion from non-DMPA-treated macaques (Table 2)

### GRFT CG FDIs protect mice against HSV-2 G and HPV16 PsV.
We evaluated the anti-HSV-2 and anti-HPV properties of GRFT/CG FDIs using mouse-sized FDIs containing 0.1 mg (4 wt. %) GRFT and 0.3 mg (12 wt.%) CG (Fig. 4a, Table 1). In the murine models of HSV-2 G (Fig. 4b) and HPV16 pseudovirus (PsV) (Fig. 4c) infection[30,31], Balb/C mice are DMPA-treated prior to virus exposure (7 or 3 days before exposure to $10^4$ plaque forming units (pfu) of HSV-2 G or $8 \times 10^6$ copies of HPV16 PsV, respectively). For HPV16 PsV, mice are also administered nonoxynol-9 vaginally 6 h before challenge to expose the basement membrane and facilitate virus binding[31]. GRFT/CG FDIs were administered 4 h before virus exposure. GRFT-only and CG-only FDIs were not tested as we previously examined the contribution of each drug to protection in the HSV and HPV PsV models using gel formulations[12].

GRFT/CG FDIs protected 9 of 15 mice from challenge with a 100% lethal dose of HSV-2 G that infected 15 of 15

**Table 1 FDI composition**

| Species | FDI | GRFT | CG | Dextran 40 | Sucrose | Mannitol | HEC |
|---|---|---|---|---|---|---|---|
| Macaque | GRFT/CG | 1 mg | 3 mg | 8 mg | 2 mg | 12 mg | 0 mg |
| | CG | 0 mg | 3 mg | 8 mg | 2 mg | 12 mg | 0 mg |
| Mouse | GRFT/CG | 0.1 mg | 0.3 mg | 0.8 mg | 0.2 mg | 1.2 mg | 0 mg |
| | HEC | 0 mg | 0 mg | 0.8 mg | 0.2 mg | 1.2 mg | 0.5 mg |

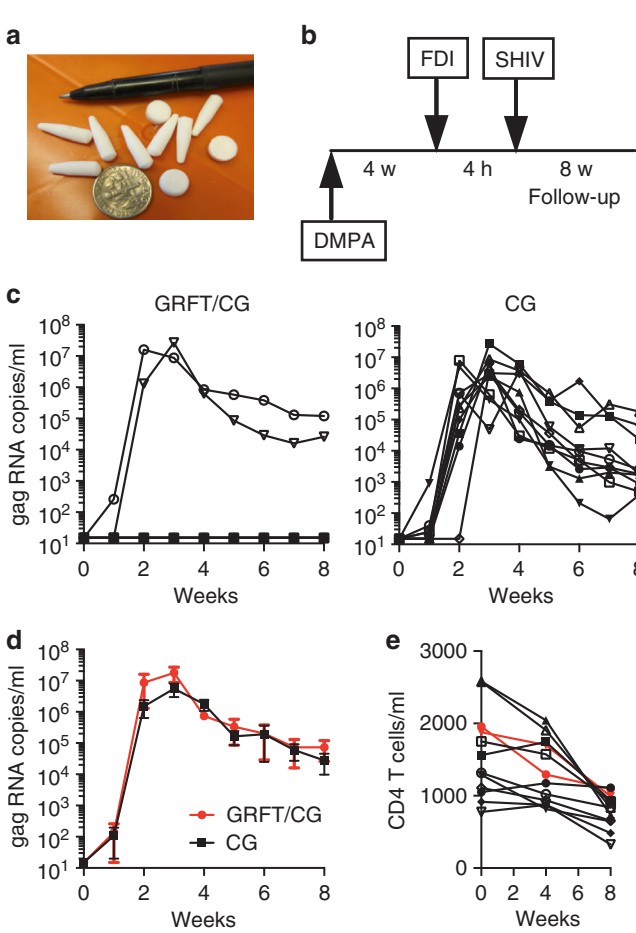

**Fig. 1** GRFT/CG FDIs protect macaques from SHIV infection. **a** Macaque sized FDIs. **b** Schematic of the macaque challenge study. **c** Plasma viral RNA copies/ml of SHIV over time following challenge in macaques administered FDIs. DMPA-treated macaques were challenged with 300 TCID$_{50}$ SHIV SF162P3 4 h after vaginal administration of either GRFT/CG FDIs (left, $n = 10$, 2 of 10 infected) or control CG FDIs containing all the same components except GRFT (right, $n = 10$, 10 of 10 infected). The percent of infected animals in each group was compared at the conclusion of the study by Fisher's Exact test. **d** The means with standard error of the mean (SEM) are shown as symbols with error bars for macaques that became infected during the study in the presence of the GRFT/CG (red symbols, $n = 2$) or CG (black symbols, $n = 10$) FDIs. **e** The number of CD4 T cells per milliliter (ml) of blood is shown over time for all macaques that became infected. Macaques exposed to GRFT/CG FDIs are in red and to CG FDIs in black

hydroxyethylcellulose (HEC) placebo FDI-treated controls (Fig. 4d). HEC was used since CG impedes HSV and HPV infections in mice[12,20,30,31]. The 63% protection vs. HEC FDI was highly significant ($p < 0.0001$, Fisher's exact test). Protection was associated with mean GRFT concentrations in mouse vaginal

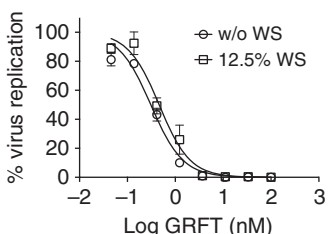

**Fig. 2** Effect of semen on GRFT anti-HIV activity. The antiviral activity of GRFT was determined using the TZM-bl MAGI assay with (open squares) or without (open circles) human whole semen (WS). The graph shows the percent of virus replication (symbols and error bars represent mean ± standard deviation (SD)) relative to virus control (triplicates per condition). The dose-response curves were used to estimate EC$_{50}$ values with 95% confidence intervals (not shown). Data are the composite of two independent experiments

washes of approximately 300-fold the anti-HSV-2 EC$_{90}$ (12 ng/ml based on GRFT concentration) (Fig. 5).

GRFT/CG FDIs also protected 10 of 10 mice against HPV16 PsV infection using in vivo luciferase expression from the reporter gene for detection and a PsV inoculum that infected 7 of 10 control HEC FDI-treated mice (Fig. 4e, f). Comparison of the log-transformed radiances revealed that protection by the GRFT/CG FDI was highly significant vs. HEC FDI ($p < 0.0001$, ANOVA with Bonferroni's correction). In contrast, radiance values in mice administered GRFT/CG FDIs were not different from background values in D-PBS treated mice that were not challenged.

**GRFT is safe and minimally absorbed after repeated use.** To further probe the safety of GRFT, we performed toxicology studies. Repeated dosing of GRFT and GRFT/CG in small animal models revealed no adverse findings at any dose levels tested, and showed that GRFT/CG gel is non-irritating (Table 3). 7 days of daily vaginal application of 0.1% GRFT/CG gel did not enhance the susceptibility of mice to HSV-2 infection when compared to the D-PBS control ($p = 0.7152$, Fisher's exact test). 14 days of daily intravenous administration of GRFT up to 8.3 mg/kg/day in rats resulted in no detectable anti-drug-antibodies (ADA) and a no adverse effect level (NOAEL) of 8.3 mg/kg/day despite high systemic levels of GRFT. Fourteen days of daily vaginal GRFT/CG gel dosing (up to 0.3% GRFT) in rats resulted in a NOAEL of 0.3% GRFT and little or no vaginal irritation. This regimen also resulted in little or no systemic detection of GRFT. A related study in rabbits also found a NOAEL of 0.3% GRFT and little or no vaginal irritation.

## Discussion

In the era of PrEP and efficacious microbicides, reducing HIV seroconversion rates remains a challenge. Young women were the least adherent to daily PrEP and the dapivirine vaginal ring in large phase 3 clinical trials[32–34]. Obtaining parental consent for PrEP or ARV-based microbicide prescriptions may be a high barrier to uptake. Availability of safe, non-ARV microbicides,

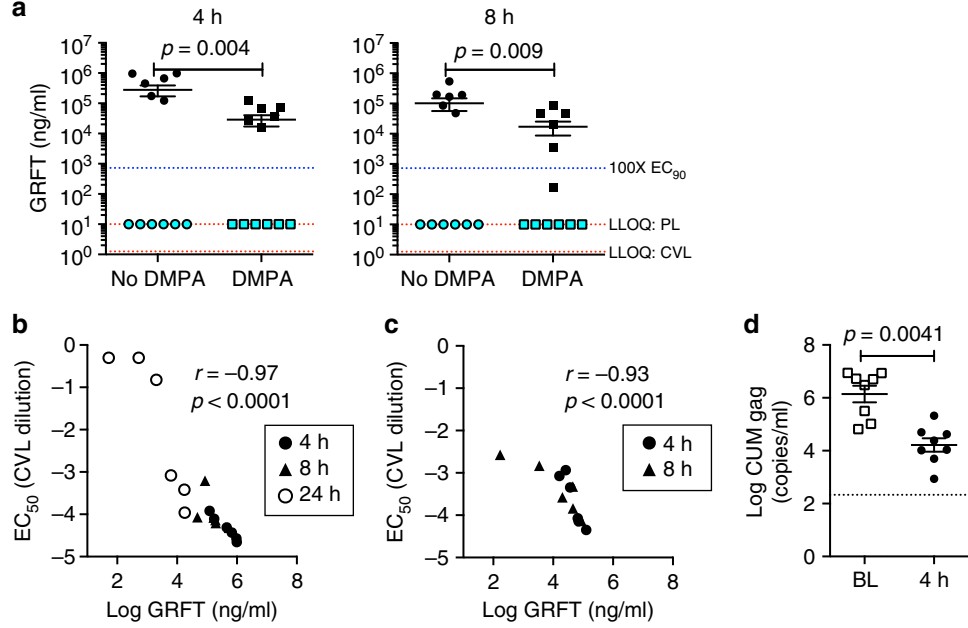

**Fig. 3** GRFT in vivo release and activity. **a** For PK evaluation, GRT/CG FDIs were inserted vaginally in macaques either treated or not 4 weeks prior with DMPA. GRFT concentrations are shown in CVL (black symbols) and plasma (aqua symbols) at 4 h (left) and 8 h (right) post-insertion. Separate groups of macaques were assayed at each time point. The mean ± SEM is indicated by line and error bars for each group of 6 macaques. GRFT was not detected in plasma (PL) above the lower limit of quantification (LLOQ) of the assay, which was 10 ng/ml (top red dotted line), and so the values are shown at the LLOQ. The LLOQ for CVL was 1.25 ng/ml (bottom red dotted line). The 100-fold $EC_{90}$ level, 724.4 ng/ml, is also indicated (blue dotted line). Concentrations of GRFT in CVL from DMPA-treated and non-DMPA-treated macaques and between 4 and 8 h in DMPA-treated macaques were compared by two-sided Mann–Whitney test and $p$ values are shown for $\alpha < 0.05$. **b** GRFT concentrations in CVLs from non-DMPA-treated macaques and (**c**) from DMPA-treated macaques correlated with the $EC_{50}$ of the CVLs using Spearman correlation analysis. Spearman correlation coefficient ($r$) and significance of the association ($p$ value) are shown. **d** Anti-SHIV SF162P3 activity of CVLs from non-DMPA-treated macaques was analyzed in human ectocervical explants. Tissue infection level (CUM SIV gag copies/ml) was compared between the Baseline (BL) and 4 h (4 h) post insertion using a log-normal mixed effects two-sided ANOVA model with time points and animal IDs nested within the experiment assumed as fixed and random effects, respectively. Four CVLs selected at random from the 6 macaques per time point were each tested twice for 8 replicates total. CVLs were collected from the same macaques at baseline and 4 h post-insertion. Mean ± SEM is indicated for each group by line and error bars

---

### Table 2 GRFT/CG FDI-induced changes in vaginal cytokines and chemokines in DMPA-treated and untreated macaques

| DMPA | | | No DMPA | | | | |
|---|---|---|---|---|---|---|---|
| Analyte | 4 h | 8 h | Analyte | 1 h | 4 h | 8 h | 24 h |
| CCL5 | nd | 0.03↓ | FGF | nd | nd | nd | 0.03↓ |
| IL-15 | nd | 0.03↓ | CCL3 | nd | nd | nd | 0.03↓ |
| MIF | 0.03↓ | 0.03↓ | CCL2 | 0.03↑ | nd | nd | nd |
| IL-1RA | nd | 0.03↓ | IL-15 | nd | nd | nd | 0.03↓ |
| CXCL10 | nd | 0.03↓ | HGF | nd | nd | nd | 0.03↓ |
| CXCL9 | nd | 0.03↓ | CXCL10 | nd | nd | nd | 0.03↓ |
| | | | CXCL9 | nd | nd | nd | 0.03↓ |

For DMPA-treated macaques, analytes not detected in CVL were: G-CSF, IL-12, CCL11, IL-17, CCL3, GM-CSF, CCL4, CCL2, EGF, IL-5, HGF, CCL22, CXCL11, TNF-α, and IL-4. Analytes unaffected by GRFT/CG FDIs at any time point vs. baseline were: FGF, IL-1β, IL-10, IL-6, VEGF, IFN-γ, IL-2, and CXCL8

For DMPA-untreated macaques, analytes not detected in CVL were: IL-10, CCL11, IL-17, GM-CSF, CCL4, IL-5, CCL22, and IL-4. Analytes unaffected by GRFT/CG FDIs at any time point vs. baseline were: IL-1β, G-CSF, IL-6, IL-12, EGF, VEGF, CXCL11, MIF, TNF-α, IFN-γ, IL-1RA, CCL5, CXCL8, and IL-2

Levels of each analyte were measured in CVL at baseline and at one of the time points post-insertion. Effects of each GRFT/CG FDI formulation at each time point were compared to baseline using two-tailed Wilcoxon Signed Rank test, $\alpha < 0.05$ and the significance of the difference ($p$-value) is reported. 'nd' indicates no significant difference

---

possibly obtainable without a prescription, could extend coverage to the most vulnerable populations. A safe and effective on-demand product that prevents HSV-2 and HPV infections along with HIV could incentivize use, improve adherence, and decrease HIV incidence. GRFT, which is a potent, broad-spectrum, poorly

absorbed, and safe non-ARV lectin, is well suited to be an on-demand multipurpose microbicide. However, the sensitivity of GRFT to oxidation under mild conditions in vitro has presented regulatory challenges for its development as a microbicide. By formulating GRFT in a low moisture content FDI (~1%), we are able to suppress oxidation compared with the aqueous gel formulation and thereby generate a stable product[27]. Here we demonstrate that GRFT/CG FDIs prevent SHIV infection in macaques while also protecting mice from HSV-2 and HPV. We further provide the requisite safety data to progress the GRFT/CG FDI into clinical testing.

GRFT prevented SHIV infection in a highly stringent SHIV macaque model designed to result in 100% infection of placebo-treated macaques after a single challenge. We used a viral inoculum far in excess of the quantities of HIV found in semen[35–38] and also employed DMPA to increase susceptibility to infection[39]. All controls did become infected after a single challenge and experienced robust infection with high peak viremia. We evaluated protection in this model 4 h post-dosing because high concentrations of GRFT were detected in CVLs at this time, and the CVLs significantly prevented SHIV infection in mucosal target cells. However, vaginal concentrations of GRFT far exceeded the in vitro effective doses within 0.5–1 h of insertion (the earliest times examined), and it is likely that high GRFT concentrations would have been detected at even earlier times. GRFT/CG FDIs dissolve in under 60 s in biologically relevant volumes of vaginal simulant in vitro[27] though in vivo dissolution times remain to be assessed. GRFT levels also remained high for

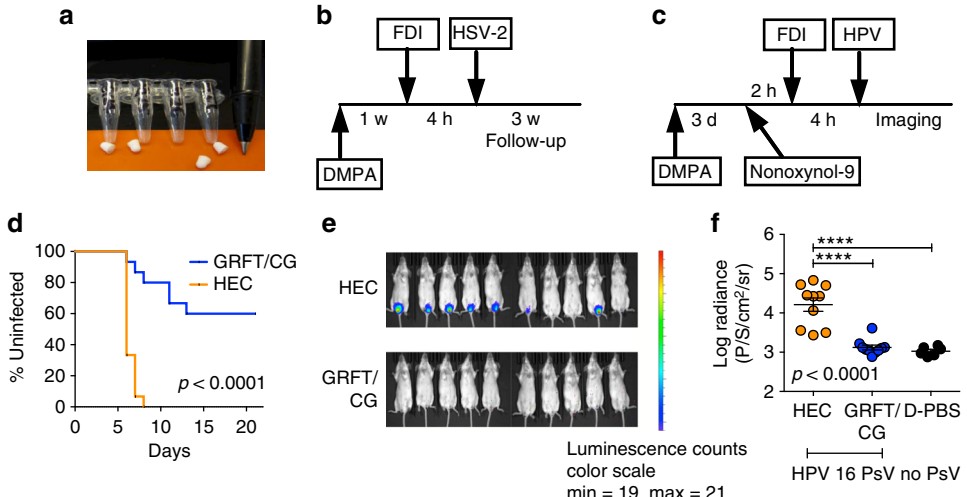

**Fig. 4** GRFT/CG FDIs protect mice from HSV-2 and HPV PsV infections. **a** Mouse-sized FDI. Schematics of the (**b**) mouse HSV-2 G and (**c**) HPV16 PsV challenge studies. **d** Survival curves showing the proportion of mice (of 15 total per group) that remained uninfected over time after HSV-2 G challenge. The GRFT/CG FDI was compared with an FDI containing HEC (HEC FDI). Significance was assessed using Fisher's exact test for the proportion infected vs. uninfected in each group at study termination. The Fisher's $p$ value is shown. **e** In vivo imaging of luminescence from HPV16 PsV challenge. Each mouse corresponds to a data point in **f**. **f** DMPA-treated mice were given one of the indicated formulations (HEC FDI, GRFT/CG FDI, or D-PBS) intravaginally 4 h before HPV 16 PsV (or no PsV for D-PBS-treated mice) challenge ($n = 10$/treatment). In vivo luciferase expression (from **e**) is expressed as mean luminescence in photons per second per centimeter squared per steridian ± SD for each individual animal. Two-sided ANOVA was used to analyze the log-transformed radiances across treatments in the HPV PsV mouse model. The F test was used for overall comparison between treatments ($p$ value in italics), and pairwise comparisons were performed using Tukey–Kramer adjusted $t$ tests with significance indicated by asterisks: ****$p < 0.0001$. Mean ± SEM is indicated for each group by line and error bars

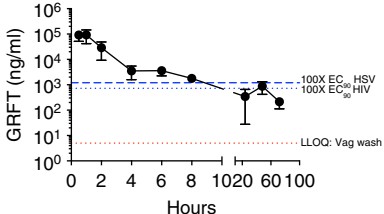

**Fig. 5** PK of GRFT in vaginal washes of DMPA-treated mice. Vaginal washes were collected from 5–6 mice per time point at 0.5, 1, 2, 4, 6, 8, 24, 48, and 72 h after FDI insertion. Separate mice were measured at each time point. The levels of GRFT equivalent to 100-fold above the anti-HIV $EC_{90}$ and 100-fold above the anti-HSV-2 $EC_{90}$ are both shown (blue dotted lines) as is the LLOQ of GRFT (red dotted line) in mouse vaginal washes (5 ng/ml). Each symbol with error bars indicates the mean ± SEM of the mice in that group

at least 8 h following FDI insertion, and GRFT was detected in the vaginal fluids of many animals even at the latest times sampled after dosing, 24 (macaques) to 72 (mice) hours. These levels were still more than 100-fold the anti-HIV $EC_{90}$ in most animals at 24 h post-dosing and more than 1000-fold in some. The window of protection could extend from within 4 min of to 8 h or more after insertion. GRFT can bind selectively to the cervical epithelium and remain on the surface of cells[6], supporting the rationale to evaluate protection in vivo at extended times after GRFT dosing. GRFT concentrations in CVLs were lower after DMPA treatment. This could be related to differences in vaginal fluid volume or viscosity, greater adherence of GRFT to the epithelium, or GRFT loss. Such differences were not observed for tenofovir administered to pigtailed macaques[40] but could reflect differences in the physicochemical properties of the drugs, drug delivery systems, or subspecies. Importantly, efficacy of the GRFT/CG FDI under non-DMPA conditions may also be even greater than we observed herein.

The two macaques that became infected in the presence of GRFT/CG FDIs were similar to uninfected macaques in characteristics that could impact vaginal distribution of GRFT or susceptibility to SHIV infection (e.g., weight, age, parity, menstrual cycle phase, and protective MHC alleles). Because we did not collect CVLs at the time of virus challenge (to not disturb the vaginal microenvironment or deplete drug), we lack other information that could have impacted SHIV transmission, such as vaginal GRFT concentration, microbiome, cytokine/chemokine milieu, and local ulceration. Because GRFT was not absorbed, there are no blood levels to indicate GRFT release from the FDI. Vaginal fluid levels and viscosity could have varied between macaques as in women[41], and GRFT could have been bound to vaginal fluid proteins[42,43].

The SHIV SF162P3 stock used for in vivo challenge herein had a low passage history and retained the full suite of potential N-linked glycosylation sites (PNGs) in gp120 present on the published SHIV SF162P3 envelope[44] less one at position 228 (N228D, N230D by HIV-1$_{HxB2}$ numbering). Loss of this PNG (N230Q) from HIV-1$_{NL4-3}$ was independently linked with increased infectivity and transmissibility of the virus[45]. This may help to explain the 100% infection rate observed in vivo with the virus stock. Loss of N230 in combination with other PNGs was also associated with increased GRFT resistance in four clade C isolates[46]. Thus, if N228D impacted GRFT activity in macaques, it would have skewed towards the null hypothesis while we saw highly significant, potent protection. Variation within the stock virus gp120 was detected around the V3 loop, and selection for a specific V3 loop sequence appeared evident in both of the infected macaques for which we could sequence envelope (one exposed to GRFT, one not). However, no differences in the two macaques' virus were detected that could be attributed to GRFT. While it is unlikely that the two infections in GRFT-exposed macaques were initiated by a GRFT-resistant viral variant in the inoculum, additional sequencing of the infected macaques will be needed to be certain. Resistance is not considered a major problem for on-

**Table 3 Safety studies of GRFT and GRFT/CG**

| Test product(s) | Route of administration, duration of dosing, animal model | Results |
|---|---|---|
| GRFT/CG gel*<br>0.1% gel | Vaginal<br>7-day repeat dosing<br>Mice | 1) GRFT/CG gel did not enhance the susceptibility of mice to HSV-2 infection when compared to D-PBS control ($p = 0.7152$, Fisher's exact test) |
| GRFT<br>2.1, 4.2, and 8.3 mg/kg/day | IV<br>14-day repeat dosing<br>Rats with TK and ADA assay | 2) NOAEL estimated to be 8.3 mg/kg/day<br>3) All blood samples tested were negative for ADA (mean sensitivity = 0.273 µg/ml for anti-GRFT antibodies) |
| GRFT/CG gel*<br>0.1%, 0.2%, and 0.3% GRFT | Vaginal<br>14-day repeat dosing<br>Rats with TK | 1) NOAEL = 0.3% GRFT/CG gel<br>2) Mean vaginal irritation scores were in the minimal to non response category for all parameters evaluated in proximal, mid-areas, and distal areas<br>3) 5 of 170 serum samples tested contained GRFT (LLOQ = 10 ng/ml)<br>4) Highest serum GRFT level (212 ng/ml) was 164-fold lower than highest Cmax value (34.7 µg/ml) seen in the IV repeat-dose study in the NOAEL group |
| GRFT/CG gel*<br>0.1%, 0.2%, and 0.3% gel | Vaginal<br>14-day repeat dosing<br>Rabbits | 1) NOAEL = 0.3% GRFT/CG gel<br>2) Mean vaginal irritation scores were in the minimal to non response category for all parameters evaluated in proximal, mid-areas, and distal areas |

*Note: 0.1% GRFT/CG gel contained 90.64% (w/w) water, 0.26% sodium acetate trihydrate, 0.35% sodium chloride, 3/1% CG, 0.20% methylparaben, 5.45% GRFT solution in PBS*
ADA antidrug antibody assay, TK toxicokinetics, NOAEL no observable adverse effect level
*3% CG gel

demand products, especially drugs like GRFT that are not absorbed after topical administration. The development of resistance to GRFT in vitro is slow, even for clade C virus under enhanced drug pressure conditions[46]. Importantly, the mechanism of action of GRFT differs from those of ARVs currently used in prevention and treatment.

We evaluated activity in single virus models (SHIV, HSV, HPV alone) while exposure scenarios are likely to involve multiple viruses. Mixed inoculum studies in explant models and in vivo, such as those we have used to evaluate ARV-based microbicides[47–49], will be essential to inform clinical progression of candidates that target multiple pathogens.

For entry blockers to impact the HIV epidemic, they must be able to act on diverse envelopes, especially those from clade C, which accounts for approximately 46% of HIV infections worldwide and dominates the epidemic in southern Africa. These agents must also be efficacious in the presence of semen. We tested the efficacy of GRFT against SHIV SF162P3, a virus with a clade B-derived envelope against which GRFT has a known low EC$_{50}$ (1.04 ng/ml). However, published data show that GRFT potently inhibits infection with transmitted HIV isolates from clade C, as well as B and also possesses activity against isolates from clade A[11]. We have found herein and in previous work that neither GRFT's anti-HIV activity nor CG's anti-HPV activity is inhibited by seminal fluids[20]. Other studies found that GRFT's anti-HSV-2 activity is preserved in the presence of seminal plasma[7]. Thus GRFT has the potential to reduce HIV transmission in the epidemic's hot spots and in the context of intercourse.

Non-ARV microbicides that have advanced into the clinic have not proven potent enough to demonstrate protection in human trials. Those that have shown the greatest efficacy in macaques (i.e., cyanovirin[3,4], glycerol monolaurate[50]) are immunomodulatory, bringing their own risks. In fact cyanovirin triggers expression of many genes and secretion of pro-inflammatory proteins[6]. GRFT, which does not induce such changes, is orders of magnitude more potent than other non-ARVs in vitro against HIV and is just as, or more potent than, ARVs. Furthermore, the benefits of combining HIV protection with HSV and HPV protection cannot be overstated[18]. We recently initiated the first-in-human Phase 1 clinical study of GRFT administered as a vaginal GRFT/CG gel (NCT02875119). The results reported here, when combined with the anticipated safety of the GRFT/CG gel in

women, should facilitate rapid progression of GRFT/CG FDIs into clinical testing to address widespread unmet needs of women globally.

## Methods

**Fast dissolving inserts.** GRFT was produced in *Nicotiana benthamiana* by infiltrating a recombinant GRFT-*Agrobacterium*, instead of a TMV vector, into *N. benthamiana* seedlings[11]. GRFT was isolated from infected leaf biomass 7 days later and purified by ion exchange chromatography. CG was obtained from Gelymar (Santiago, Chile). GRFT and CG solutions were combined with excipients (Table 1), loaded into polymerase chain reaction (PCR) tubes (Agilent Technologies, Santa Clara, CA), and placed in a Millrock Laboratory Freeze Dryer (LD85, Millrock Technology, Kingston, NY) with condenser temperature −70 °C and vacuum 100 mTorr. Formulations were rapidly frozen to −45 °C and held for 3 h[27]. Primary drying was performed from −40 °C to 30 °C over 22 h, and secondary drying was at 30 °C for 5 h followed by a 4 °C hold. PCR tubes were capped and sealed in aluminum foil sachets (Pharmaceutical Packaging Services, Richmond, VA) with a MediVac Sealer (ALINE Heat Seal Corporation, Cerritos, CA) and stored at 2–8 °C until FDIs were popped out of the tubes for use. Given the stability of human-sized GRFT/CG FDIs prepared in aluminum foil-sealed blister sheets up to 40 °C/75% relative humidity[27], FDIs for commercial use could be stored at ambient temperature. FDI composition is shown in Table 1.

**TZM-bl MAGI assay with semen.** The multinuclear activation of a galactosidase indicator (MAGI) assay in TZM-bl cells (NIH AIDS Reagent Program)[24] was modified for evaluation of semen effects as follows: TMZ-bl cells were pre-incubated for 15 min with different concentrations of GRFT. Cell-free virus (HIV-1$_{ADA-M}$) was diluted with medium containing 25% whole human semen and the resulting mixture was added to TZM-bl cells for a final semen concentration of 12.5%. A similar antiviral assay without human semen was performed side-by-side as control. The TCID$_{50}$ and 95% CI were calculated using a curve-fitting analysis with GraphPad Prism (La Jolla, CA).

**Anti-SHIV activity in macaques.** Adult female Indian rhesus macaques (*Macaca mulatta*) that tested negative by serology for simian retrovirus, Herpes B, simian T cell leukemia virus type 1, and SIV were enrolled. Macaque studies were carried out at Tulane National Primate Research Center (TNPRC, Covington, LA) in compliance with the regulations stated in the Animal Welfare Act, the Guide for the Care and Use of Laboratory Animals, and TNPRC animal care procedures[51,52]. The TNPRC Institutional Animal Care and Use Committee (IACUC) approved the studies (OLAW Assurance #A4499-01). TNPRC receives full accreditation by the Association for Accreditation of Laboratory Animal Care (AAALAC #000594). Animals were socially housed indoors in climate-controlled conditions and monitored twice daily by a team of veterinarians and technicians to ensure the animals' welfare. Any abnormalities were recorded and reported to a veterinarian. Macaques were fed commercially prepared monkey chow twice daily along with supplemental foods including fruit, vegetables, and foraging treats as part of TNPRC's environmental enrichment program. Water was available continuously. TNPRC Division of Veterinary Medicine has established procedures to minimize pain and

distress through several means in accordance with the Weatherall Report. Before all procedures, including blood collection, macaques were anesthetized with ketamine-HCl (10 mg/kg) or tiletamine/zolazepam (6 mg/kg). Preemptive and post-procedural analgesia (buprenorphine 0.01 mg/kg) was administered for procedures that could cause more than momentary pain or distress in humans undergoing the same procedures. All macaques were released, not euthanized, at the conclusion of the study.

SHIV SF162P3 stock used for in vivo challenge was a third-generation growth of virus obtained originally from the NIH Division of AIDS. The stock was grown in CD8 T cell-depleted allogeneic rhesus macaque peripheral blood mononuclear cells (PBMCs) as follows: The cells ($10^7$/ml) were stimulated for 3 days with 3 µg/ml phytohemagglutinin (PHA) and 40 U/ml IL-2, washed, and infected with a SHIV SF162P3 stock grown in the lab (200 $TCID_{50}$ per $10^6$ cells). This parent stock was a second-generation growth from the Division of AIDS stock. The following day and every 3–4 days thereafter, feeder PBMCs were added in media with IL-2. Cultures were maintained at $5 \times 10^6$ cells/ml for 10 days, and supernatants were collected, clarified by centrifugation, and frozen at −80 °C. Virus growth was monitored by p27 enzyme linked immunosorbent assay (ELISA). The harvested virus stock was characterized for p27 content by the ELISA, SIV gag RNA content by quantitative reverse transcription PCR (qRT-PCR)[53], $TCID_{50}$ in rhesus macaque PBMCs by Reed and Muench method[54], and focus forming units in TZM-bl cells by MAGI assay for repeated titration of the same stock over time. qRT-PCR was performed on viral RNA isolated from the stock with the Qiagen RNeasy kit and amplified by the standard curve method using the One-step RT-qPCR Kit (KAPA Biosystems, Wilmington, MA) on a ViiA-7 Real-Time PCR System (Thermo Fisher Scientific, Waltham, MA) with the following primers: SIVgag FW (5′-GGTTGCACCCCC TATGACAT-3′), SIVgag RV (5′-TGCATAG CCGCTTGATGGT-3′). SIV gag plasmid was used for the standard curve. MAGI assay was performed using HIV-1$_{ADA-M}$[24]. Serial dilutions of virus were tested in triplicate to establish the dose-response curve. The $TCID_{50}$ and 95% CI were calculated using a curve-fitting analysis with GraphPad Prism (La Jolla, CA). The titer used for in vivo infection was $TCID_{50}$ from macaque PBMCs.

Macaques were challenged intravaginally with 300 $TCID_{50}$ SHIV SF162P3 4 weeks after 30 mg intramuscular DMPA injection. 4 h before challenge, macaques had GRFT/CG or CG FDIs ($n = 10$ each) inserted intravaginally. The animals were followed for 8 weeks. At the times of FDI insertion and challenge and during follow up, blood was collected for the isolation of plasma and PBMCs by Ficoll-Hypaque (GE Healthcare, Chicago, IL) centrifugation[55]. SHIV viral load in plasma was quantified by quantitative reverse transcriptase PCR (qRT-PCR)[56]. Primers and probe were SGAG21 (forward), 5′-GTC TGC GTC ATP TGG TGC ATT C-3′; SGAG22 (reverse), 5′-CAC TAG KTG TCT CTG CAC TAT PTG TTT TG-3′; and pSGAG23 (probe, 100 nM), 5′- (FAM) CTT CPT CAG TKT GTT TAC TTT CTC TTC TGC G-(BHQTM1)-3′. The lower limit of quantification of the assay was 15 RNA copies/ml.

For PK measurements, macaques were administered FDIs intravaginally, and blood and CVLs were collected at 1, 4, 8, or 24 h post-insertion ($n = 6$ macaques/time point). GRFT was measured in plasma and CVLs. Nine months later, macaques were injected intramuscularly with 30 mg DMPA and administered FDIs again 4 weeks post-DMPA. Blood and CVLs were collected at 4 or 8 h post-insertion for GRFT measurement.

Anti-HIV activity in CVLs from macaques in PK studies was assessed by TZM-bl MAGI assay with HIV-1$_{ADA-M}$ as described above[24]. Serial dilutions of CVL were tested in triplicate to establish the dose-response curve. The $EC_{50}$ and 95% CI were calculated by curve-fitting analysis.

Anti-SHIV SF162P3 activity of macaque CVLs in mucosal target cells was tested in human ectocervical tissues without gross pathological changes from women undergoing routine hysterectomy. Tissues were received from the National Disease Research Interchange (NDRI, Philadelphia, PA) and processed for polarized explant cultures ($5 \times 5$ mm; 1-2 explants per condition)[57]. After 48 h of activation with 5 µg/ml PHA and 100 U/ml interleukin-2 (IL-2), explants were challenged with 60 $TCID_{50}$ of SHIV SF162P3 (20 µl) mixed with CVL (20 µl) applied on the apical surface of the epithelium for 4 h. Tissues were washed extensively and cultured for 14 days. Infection was monitored by SIV gag qRT-PCR[47] performed on culture supernatants collected every 3–4 days and analyzed for SOFT and CUM endpoints[47,58–60]. The CUM was reported. Activity of each CVL pair (baseline vs. 4 hour, $n = 4$) was tested twice in separate experiments.

Vaginal pH and cytokines and chemokines were quantified to assess vaginal safety of GRFT/CG FDIs. Vaginal pH was measured with litmus paper inserted into the vaginal vault for 5 min. 29-plex Luminex quantified cytokines and chemokines in CVL. The Novex® Monkey Cytokine Magnetic 29-Plex Panel kit (Life Technologies, Carlsbad, CA) was used on a MAGPIX® system (Luminex XMAP Technology, Austin, TX) with Luminex xPOPNENT software. Clarified macaque CVL supernatants were thawed, centrifuged, and aliquotted for 1:3 final dilution. The assay was performed according to the manufacturer's instructions. Values that fell within the standard curve for each analyte were plotted. Values below the lowest standard concentration were plotted as the lowest standard concentration.

**Anti-HSV-2 and HPV16 PsV activity in mice.** Studies using female Balb/C mice were carried out at Rockefeller University's Comparative Bioscience Center (RU

CBC, New York, NY) following the guidelines of the Animal Welfare Act and the Guide for the Care and Use of Laboratory Animals[51,52]. RU IACUC approved the animal protocols (protocol numbers 12563 and 14684-H). Veterinarians at CBC regularly monitored the animals to minimize any distress or pain.

HSV-2 G (ATCC) was propagated in Vero cells (ATCC), and the titer was determined by plaque assay[30]. To test antiviral activity against HSV-2, we performed vaginal challenge with $10^4$ pfu/mouse HSV-2 G in mice pre-treated with 2.5 mg DMPA[31]. Mice were scored daily from day 4 to to day 25 post-challenge. Animals with signs of infection (e.g., hind limb paralysis, erythema, hair loss, and vaginal swelling) were deemed infected and euthanized by carbon dioxide inhalation. HPV16 PsV was produced by co-transfection of 293 T cells (NCI, Frederick) with p16shell and Addgene (Cambridge, MA) plasmid 37328 (reporter pCLucf)[31]. Titration was performed by qRT-PCR for the reporter using the Absolute Blue qPCR Sybr Green kit (Thermo Fisher) with the following primers for EGFP: Forward (5′-GAG CTG AAG GGC ATC GAC TT-3′) and Reverse (5′-CTT GTG CCC CAG GAT GTT G-3′). Reactions were run on the Viaa7. We performed vaginal high-dose challenge with $8 \times 10^6$ copies in 10 µl HPV16 PsV in mice pre-treated with DMPA/nonoxynol-9 (Fig. 3b)[20,22,31]. We measured luciferase expression 24 h after challenge by vaginal application of D-luciferin followed by imaging on an IVIS spectrum imaging system (PerkinElmer, Waltham, MA)[31].

For PK measurements, DMPA-treated mice ($n = 6$/time point) were administered FDIs intravaginally, and vaginal washes were collected[20] at 0.5, 1, 2, 4, 6, 8, 24, 48, and 72 h post FDI insertion.

**GRFT ELISA.** GRFT was quantified using a validated indirect sandwich ELISA. 96-well plates were pre-coated with HIV-1$_{BaL}$ gp-120 (NIH Reagent Program Cat#49610, Germantown, MD) overnight at 4 °C. Wells were blocked with 0.05% ovalbumin (Sigma, St. Louis, MO), 0.1% Tween 20 (Sigma) in PBS (Sigma) at 37 °C for 1.5 h. Standards, controls and samples were pipetted in duplicate into the wells and incubated at 37 °C for 1 h. A goat anti-GRFT detection antibody (0.5 µg/ml, Pacific Immunology, Ramona, CA) was added for 1 h at 37 °C, followed by a rabbit anti-goat-HRP secondary antibody (0.2 µg/ml, Southern Biotech, Birmingham, AL) incubated for 30 min at 37 °C. Ultra-TMB substrate (Thermo Scientific, Rockford, IL) was added followed by 0.16 M sulfuric acid (Thermo Scientific). Plates were washed with 0.1% Tween 20/PBS between each step. Plates were read on the Emax microplate reader (Molecular Devices, Sunnyvale, CA) using 450 nm for absorbance and 570 nm for reference. The lower limits of quantification were 1.25, 5, and 10 ng/ml for macaque CVLs, macaque plasma and mouse vaginal washes, respectively.

**Repeated dosing safety and toxicology.** GRFT/CG gels were prepared as follows: Sterile filtered water, sodium acetate trihydrate, and sodium chloride were mixed and heated to 69 °C. CG was added with stirring for 3.5 h and cooled to 60 °C for addition of methyl paraben in water. Following 30 min of stirring, the mixture was further cooled to 21 °C and stirred 45 min longer. GRFT in PBS (for final 0.1%, 0.2%, or 0.3% gels) was then added at RT and stirred for 30 min. The gels were characterized for pH, viscosity, GRFT content, methyl paraben content, osmolality, and turbidity. Gels were stored at 4 °C until use.

The HSV-2 infection enhancement model was performed as follows: GRFT/CG gel containing 0.1% GRFT (10 µl) was administered vaginally to Balb/C mice daily for 7 days. Following the established protocol[30], the mice were challenged 12 h after the last gel application with a suboptimal inoculum of $2 \times 10^3$ pfu HSV-2 G that infects only 50% of control D-PCS-treated mice. Beginning on day 4 and for 21 days total, mice were scored for signs of infection.

Intravenous toxicity was assessed as follows: GRFT was dosed intravenously daily for 14 days in 6 week old male and female Sprague-Dawley rats followed by a 14-day observation period following the final dose. Dose levels evaluated were 2.1, 4.15, and 8.3 mg/kg/day. Toxicokentic assessment was conducted, as was anti-drug antibody (ADA) testing. Toxicokinetic assessment included cage-side and detailed clinical observations, body weights, food consumption, clinical labs, organ weights, and macroscopic and microscopic pathology. ADA testing was performed by a validated assay developed and carried out at MPI Research (Mattawan, MI) as follows: ELISA plates were coated with GRFT, blocked in 5% bovine serum albumin (BSA) in PBS, incubated, and washed. Goat-anti-GRFT positive control (Pacific Immunology, Ramona, CA) and dilutions of the serum samples were added to the plates in duplicate, and the plates were incubated and washed. Peroxidase-conjugated AffiniPure bovine anti-goat IgG (Jackson ImmunoResearch, West Grove, PA) was added, and the plates were incubated and washed. 1-Step$^{TM}$ Ultra TMB-ELISA substrate was added, and the plates were incubated. Reaction development was stopped with 2 N sulfuric acid, the optical density (OD) measured at 450 nm, and the data analyzed with SOFTmax Pro GxP Version 5.3. The assay had a mean sensitivity of 0.273 µg/ml for anti-GRFT antibodies.

Vaginal toxicity was assessed as follows: GRFT/CG gels containing 0.1% (1 mg/ml), 0.2% (2 mg/ml), or 0.3% (3 mg/ml) GRFT were dosed vaginally in 6 week old Sprague Dawley rats daily for 14 consecutive days followed by a 14-day recovery period. Systemic exposure to GRFT was measured by ELISA as described in the Methods (LLOQ for rat serum was 2.5 ng/ml in a minimum required dilution of 1:4, corresponding to blood levels of 10 ng/ml). Toxicokinetics were evaluated as in the intravenous dosing study.

GRFT/CG gels containing 0.1%, 0.2%, or 0.3% GRFT were also dosed vaginally in 5–8 months old New Zealand white rabbits for 14 consecutive days followed by a 14-day recovery period. Toxicokinetics were evaluated as in the rat intravenous and vaginal dosing studies. In addition, vaginal irritation scoring was performed on the proximal, mid-, and distal vaginal areas.

**Statistics**. GraphPad Prism 5.02 and SAS (Cary, NC) were used to analyze the data, which were graphed in Prism. Macaque data were analyzed using non-parametric tests or were log-transformed and analyzed using parametric tests. Mouse data were analyzed using parametric tests. Multiple comparison corrections were applied where appropriate. All statistical tests are indicated in the relevant sections of Results and within the Figure Legends.

## Data availability

All data are available from the authors. The 16 SHIV SF162P3 sequences derived in these studies have been deposited in GenBank and have the following accession codes: MH716498, MH716499, MH716500, MH716501, MH716502, MH716503, MH716504, MH716505, MH716506, MH716507, MH716508, MH716509, MH716510, MH716511, MH716512, MH716513.

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

## Acknowledgements

We thank the veterinary staff at the TNPRC and the CBC at RU for continued support and Randy Fast, Kelli Oswald and Rebecca Shoemaker of the Quantitative Molecular Diagnostics Core of the AIDS and Cancer Virus Program of the Frederick National Laboratory for Cancer Research for expert assistance with plasma viral load measurements. The funders had no role in study design, data collection, and analysis, decision to publish, or preparation of the manuscript. The findings and conclusions in this report are ours and do not necessarily represent the views of the Centers for Disease Control and Prevention. This research has been supported by the President's Emergency Plan for AIDS Relief (PEPFAR) through the U.S. Agency for International Development (USAID) under the terms of GPO-A-00-04-00019-00, and in part with funds from the National Cancer Institute, NIH, under contract HHSN261200800001E. This study was supported in part by the TNPRC Base Grant P51OD011104-56.

## Author contributions

N.D. designed and oversaw the SHIV macaque studies and sequencing, analyzed the data, wrote, and revised the paper. M.Lal. and M.Lai. designed, manufactured, and characterized the FDIs. M.A., P.B., and O.M. processed samples from macaque studies and performed in vitro assays. L.K., A.R., and K.L. performed mouse experiments and quantified GRFT in PK samples. A.W. and S.U. characterized the FDIs. K.K. coordinated toxicology studies. M.M.P., Z.L., and M.B. performed toxicology studies. J.D.L. carried out SHIV viral load testing and critically reviewed the paper. W.H., B.O.K., E.M., and M.R. contributed to study design and critically reviewed the paper. B.G., J.B., and A.G. carried out the macaque studies. N.T. designed and oversaw the macaque PK/PD studies, analyzed the data, and assisted in writing the paper. J.A.F.R. designed and oversaw the mouse experiments and GRFT quantification, analyzed the data, and assisted in writing the paper. T.M.Z. oversaw the project and assisted in writing the paper.

## Additional information

**Competing interests:** The authors declare no competing interests.

