## [Peer Review File · Nature Communications]

Reviewers' Comments:

Reviewer #1:

Remarks to the Author:

The manuscript by Nina Derby and colleagues details in vivo studies with fast dissolving vaginal inserts (FDI) comprised of Carrageenan (CG) and the antiviral lectin Griffithsin (GRFT). Their data convincingly show that the CG/GRFT FDI protect female rhesus macaques from high dose challenge with Simian-Human Immunodeficiency virus, providing strong support for this product concept to be used as an on-demand topical pre-exposure prophylaxis strategy. These exciting data are further supported by data showing that similar formulations protected mice against genital herpes (HSV-2) challenge, and human papillomavirus type 16 pseudovirus challenge, although the novelty of the latter report is not high given that the same group has shown that CG/GRFT gel formulations also have protective efficacy in the same preclinical models. I believe that these data provide strong support for moving Griffithsin-based products such as the authors FDI forward in clinical testing, but my enthusiasm for the report as written is tempered by some deficiencies in the manuscript and data discussion.

In the first instance, the manuscript is not very well written. The opening paragraph attempts to provide a compelling justification for non-antiretroviral based topical pre-exposure prophylaxis, but the sentences are long and convoluted. This continues throughout the text, and I suggest the manuscript would benefit from careful proofreading.

I find the repeated referencing of "unpublished data" inappropriate. For example citation on page 7 that GRFT has an excellent safety profile is supported by reference 6: Zydowsky T, unpublished. An important published safety metric for GRFT is not cited at the appropriate place at the end of the first sentence of the 3rd paragraph or the introduction: Nixon et al. showed that GRFT did not increase susceptibility to HSV-2 challenge. (reference 12),. Citation that GRFT and CG activity is inhibited by seminal fluids is supported by reference 39: Fernandez Romero JA unpublished (page 8). Reference 23 is to a manuscript under review. Reference 14 is to a presentation that is not publically available (no DOI). Reference 18 is to a conference presentation, likewise not publically available. Reference 5 is not the primary reference for safety profile of CV-N – this should be Huskens et al. *Int J Biochem Cell Biol.* 2008;40(12):2802-14. doi: 10.1016/. The statement regarding HIV/HSV/HPV "intertwined epidemiology" requires supportive citation.

The efficacy data are convincing and exciting, but I would like to see some explanation for why the GRFT accumulation levels are lower in DMPA treated animals in comparison with non-DMPA treated animals. In the mouse HSV and HPV treatment studies, an important control group is left out: the experimental cohort are treated with CG-GRFT FDI, and the control group with HEC FDI because CG has inherent HSV and HPV inhibitory activity. This misses the opportunity to define the role of the GRFT incorporated in the formulation in protecting animals against these pathogens. A CG only FDI should have been tested, and a GRFT only FDI should have been tested. As presented the data cannot discern whether CG/GRFT FDI are superior to CG FDI or GRFT FDI.

In conclusion, the data have merit, but citation of supporting unpublished data is unacceptable and the manuscript does not stand on its own without them. I suggest the supporting information should be provided in a Supplementary Information section. I also suggest that the manuscript would benefit from careful proofreading.

Kenneth E. Palmer

Reviewer #2:

Remarks to the Author:

In this paper, the authors have undertaken a careful evaluation of a griffithsin (GRFT)/carrageenan

(CG)fast dissolving insert (FDI) for the prevention of SHIV, HSV-2 and HPV infections. Efficacy studies were undertaken in a non-human primate model (for SHIV efficacy) and the mouse model (for HSV-2 and HPV infection). The studies were well designed, appropriate, and conclusive and demonstrate significant efficacy for these three pathogens. The development of a multi-purpose FDI is highly responsive to the current direction of HIV prevention research. Additional data showing persistence of the antiviral efficacy out to 8 hours post dosing is also very encouraging for a pericoital product.

I have no concerns about the technical aspects of the study but do think that the paper would be strengthened by a brief discussion of what (if anything) is known about the safety and PK profile of repeated exposure to griffithsin/carrageenan FDI or gel. The authors mention that a Phase 1 gel study is ongoing and so I assume that there are repeat dosing GLP animal toxicology data. One potential concern related to the use of protein based products is the possibility that local or systemic absorption of the product might induce immunological responses that could reduce product efficacy. The authors state that single dose administration was not associated with systemic PK exposure but it would be interesting to know if there are any repeat dose data.

The authors mention that the product was stored at 2-8°C until the FDIs were used in the challenge experiments. Does this imply that the product would require commercial storage in this temperature range.

In addition, there are reports that (1) GRFT is prone to oxidation, by both hydrogen peroxide exposure and human cervicovaginal secretion exposure; (2) Methionine at position 78 in the amino acid sequence of GRFT is oxidized. The authors do not comments on the stability of the FDI to oxidation and it would be useful to have this information if it is available for the GRFT/CG FDI.

Reviewer #3:

Remarks to the Author:

In the manuscript "Griffithsin/Carrageenan Fast Dissolving Inserts Prevent SHIV, HSV-2 and HPV Infections in vivo", Derby et al present a series of experiments showing the efficacy of a new antimicrobial compound/formulation in preventing several vaginal viral infections. Here they report the effects of a new Griffithsin/Carrageenan fast dissolving insert in macaque and mouse models of SHIV, HSV-2, and HPV infections. Both compounds have been previously reported to provide some protective efficacy in vitro and in vivo against the three viruses tested here. Interestingly, the GR/CG FDIs provide drug intravaginally for 8 hours (and in some cases up to 24 hours) after insertion without altering vaginal pH and without entering circulation. The authors first present the results of an animal study where 8 out of 10 RMs were protected from a single moderate dose vaginal SHIVsf162P3 challenge. The authors then describe two mouse studies showing protection from high dose (lethal) HSV-2 challenge (63%) and HPV16 pseudovirus infection (100%).

This is a highly important, well-written study of a new antimicrobial treatment that could provide women across the globe with an effective, discreet means of protecting themselves from HIV transmission. Acknowledging space limitations, this manuscript could be improved by addressing the following comments.

Minor Comments:

1. 647 TCID50 isn't high dose viral challenge. This is a moderate dose at best. If you are comparing the dose to what you expect semen viral titers to be then make this clear and cite your sources.
2. It isn't clear from the methods section how many times this virus has been passaged in vitro since its original growth. Mutations will occur in the culture. Thus the extent to which these

mutations will have any impact on transmission is actually up in the air (refer to page 8 lines 10-11.) It shouldn't be hard to sequence envelope in your virus stock and determine whether the virus lost glycans over its passage history.

3. HPV pseudovirus didn't infect 100% of controls suggesting that the virus stock had a low titer, so is sterilization actually 100%??

4. The effectiveness of any treatment focused on entry will depend largely on its ability to neutralize multiple subtypes and multiple different envelope sequences within subtypes. Is there evidence that some envelopes may be less susceptible? Please discuss this in the manuscript.

Grammatical comments:

1. Methods incomplete: cytokine assays, menstrual cycle determination, expand (briefly) pseudovirus assay / detection.

2. Define acronyms at first reference: DMPA, CVL

3. page 9: "*Anti-SHIV activity in macaques*" in italics doesn't have a paragraph associated with it.

Point By Point Response to Reviewers' Comments

Please see below for the responses to each individual concern raised by each of the Reviewers. Please note that the reference numbers have changed as references have been added and removed throughout. Citations are referred to in this Response document by their original reference number, as well as the first author and date. Reviewer comments are provided in blue text. Our responses are provided in black. Line numbers refer to the clean version of the manuscript file with continuous numbering.

Reviewer 1: Dr. Kenneth Palmer

The manuscript by Nina Derby and colleagues details in vivo studies with fast dissolving vaginal inserts (FDI) comprised of Carageenan (CG) and the antiviral lectin Griffithsin (GRFT). Their data convincingly show that the CG/GRFT FDI protect female rhesus macaques from high dose challenge with Simian- Human Immunodeficiency virus, providing strong support for this product concept to be used as an on-demand topical pre-exposure prophylaxis strategy. These exciting data are further supported by data showing that similar formulations protected mice against genital herpes (HSV-2) challenge, and human papillomavirus type 16 pseudovirus challenge, although the novelty of the later report is not high given that the same group has shown that CG/GRFT gel formulations also have protective efficacy in the same preclinical models. I believe that these data provide strong support for moving Griffithsin- based products such as the authors FDI forward in clinical testing, but my enthusiasm ~~is~~ ^{for} the report as written is tempered by some deficiencies in the manuscript and data discussion.

In the first instance, the manuscript is not very well written. The opening paragraph attempts to provide a compelling justification for non-antiretroviral based topical pre-exposure prophylaxis, but the sentences are long and convoluted. This continues throughout the text, and I suggest the manuscript would benefit from careful proofreading.

We appreciate Dr. Palmer's careful attention to the writing style. We have carefully reviewed the entire manuscript and revised long and/or convoluted sentences. See specifically line 51-52, 74-75, 83-85, 94-95, 113-115, 134-135, 145, 151-153, 253-254.

I find the repeated referencing of "unpublished data" inappropriate.

We are happy to be able to include the requested publication references and missing data. Please see below for the specific information requested for each of the requests.

For example citation on page 7 that GRFT has an excellent safety profile is supported by reference 6: Zydowsky T, unpublished.

Given the additional interest by Reviewer 2 in the safety data, we have decided to include additional safety data within the manuscript as Supplementary Data (Supplementary Table 3) with accompanying text in a final Results section entitled "GRFT remains safe and minimally absorbed after repeated exposure". This can be found on line 156-167. The appropriate methods are in Supplementary Methods on line 587-627. (Please see response to Reviewer 2 for further discussion). Thus, the safety data are no longer referred to as unpublished.

An important published safety metric for GRFT is not cited at the appropriate place at the end of the first sentence of the 3rd paragraph or the introduction: Nixon et al. showed that GRFT did not increase susceptibility to HSV-2 challenge. (reference 12),.

We thank Dr. Palmer for suggesting we include this information here. We have added reference to Nixon, et al. 2013 in the recommended place on line 66. This is now reference 7. Please note that track changes did not capture the insertion and removal of references.

Citation that GRFT and CG activity is inhibited by seminal fluids is supported by reference 39: Fernandez Romero JA unpublished (page 8).

We have decided to include the data as a Supplementary Figure with accompanying text at the end of the first section of Results on “GRFT/CG FDIs protect from SHIV-SF162P3 vaginal infection” (line 105-106) with associated methods provided in Supplementary Methods section (line 570-575). Thus the data are no longer referred to as unpublished. We have also mentioned these data in the Discussion on line 245, 249, and 252.

Reference 23 is to a manuscript under review.

This manuscript is now published in the *Journal of Pharmaceutical Sciences* and is referenced accordingly. Lal, et al. 2018 J. Pharm Sci. Reference to this manuscript (now reference 27) is found on line 85, 110, 182, 195, 272 and 277.

Reference 14 is to a presentation that is not publically available (no DOI).

This paper was written and published by AVAC on work funded by the Bill and Melinda Gates Foundation but was never published in a peer-reviewed journal due to lack of interest (Personal communication from the senior author, Lut Van Damme, of the Gates Foundation). Although a PDF of the paper is freely accessible by searching the paper title on the internet, we have removed the reference per Dr. Palmer’s request. It is no longer a reference for the statement on line 75.

Reference 18 is to a conference presentation, likewise not publically available.

The data provided in the conference poster are now In Press as a manuscript in a peer reviewed journal (Magnan, et al 2018). That manuscript is now cited in place of the poster abstract on line 78 (now reference 25).

Reference 5 is not the primary reference for safety profile of CV-N – this should be Huskens et al. *Int J Biochem Cell Biol.* 2008;40(12):2802-14. doi: 10.1016/.

We have added reference to Huskens, et al. 2008 as requested on line 64 (now reference 5). We thank Dr. Palmer for the citation.

The statement regarding HIV/HSV/HPV “intertwined epidemiology” requires supportive citation.

Dr. Palmer correctly points out that this statement should be referenced. There is a large body of primary literature that supports that

- The epidemics of HIV, HSV, and HPV overlap globally.
- Prevalent genital HSV and HPV infections increase HIV risk.
- HIV/HSV co-infected people may experience higher HIV viral loads in blood and genital secretions and increased pathogenesis than people not infected with HSV.
- HIV/HPV co-infected people may progress faster to cervical cancer than people not infected with HIV.

Given the large number of studies, we have included on line 73 the following references to epidemiological work and reviews covering additional primary studies:

- E. Schelar, C. Polis, T. Essam, K. Looker, L. Bruni, C. Chrisman, J. Manning, Multipurpose prevention technologies for sexual and reproductive health: mapping global needs for introduction of new preventive products. *Contraception* **93**, 32-43 (2016).
- K. J. Looker, J. A. R. Elmes, S. L. Gottlieb, J. T. Schiffer, P. Vickerman, K. M. E. Turner, M. C. Boily. Effect of HSV-2 infection on subsequent HIV acquisition: an updated systematic review and meta-analysis. *Lancet Infect Dis* **17**, 1303-1316 (2017).
- C. F. Houlihan, N. L. Larke, D. Watson-Jones, K. K. Smith-McCune, S. Shiboski, P. E. Gravitt, J. S. Smith, L. Kuhn, C. Wang, R. Hayes, Human papillomavirus infection and

increased risk of HIV acquisition. A systematic review and meta-analysis. *AIDS* **26**, 2211-2222 (2012).

- P. Van de Perre, M. Segondy, V. Foulongne, A. Ouedraogo, I. Konate, J. M. Huraux, P. Mayaud, N. Nagot, Herpes simplex virus and HIV-1: deciphering viral synergy. *Lancet Infect Dis* **8**, 490-497 (2008).
- H. K. Whitham, S. E. Hawes, H. Chu, J. M. Oakes, A. R. Lifson, N. B. Kiviat, P. S. Sow, G. S. Gottlieb, S. Ba, M. P. Sy, S. L. Kulasingam, A Comparison of the Natural History of HPV Infection and Cervical Abnormalities among HIV-Positive and HIV-Negative Women in Senegal, Africa. *Cancer Epidemiol Biomarkers Prev* **26**, 886-894 (2017).
- J. A. Fernandez-Romero, C. Deal, B. C. Herold, J. Schiller, D. Patton, T. Zydowsky, J. Romano, C. D. Petro, M. Narasimhan. Multipurpose prevention technologies: the future of HIV and STI protection. *Trends Microbiol* **23**, 429-436 (2015).

The efficacy data are convincing and exciting, but I would like to see some explanation for why the GRFT accumulation levels are lower in DMPA treated animals in comparison with non-DMPA treated animals.

We did not initially include discussion of this finding because the explanations are largely speculative at this stage. However, we appreciate the importance of discussing it and have now included a few sentences on possible reasons for the difference in PK between DMPA and non-DMPA conditions in the Discussion at the end of the paragraph beginning “GRFT prevented SHIV infection in a highly stringent SHIV macaque model...” (line 203-207).

In the mouse HSV and HPV treatment studies, an important control group is left out: the experimental cohort are treated with CG-GRFT FDI, and the control group with HEC FDI because CG has inherent HSV and HPV inhibitory activity. This misses the opportunity to define the role of the GRFT incorporated in the formulation in protecting animals against these pathogens. A CG only FDI should have been tested, and a GRFT only FDI should have been tested. As presented the data cannot discern whether CG/GRFT FDI are superior to CG FDI or GRFT FDI.

We realize that we did not clearly describe our previous studies using a gel formulation of GRFT/CG (Levendosky, et al. 2015, now reference 12). In the Levendosky, et al manuscript, we performed in vitro and in vivo studies and evaluated GRFT and CG each alone and together against each pathogen as Dr. Palmer indicated was the correct controlled experiment. In the in vitro experiments, we found that combining GRFT with CG improved the antiviral activity against each virus. In the HSV mouse model, we similarly found that the combination was more effective than either CG or GRFT alone, indicating roles for both compounds in the inhibitory activity. However, in the HPV pseudovirus mouse model, the potent anti-HPV activity of CG predominated, and we were unable to detect an effect of GRFT on top of the great inhibition by CG. In the studies using FDI formulations of GRFT/CG presented in the current manuscript, we decided against repeating the additional groups in the mouse models since we had already demonstrated the result for the gel formulation. We have added a statement to this effect at the end of the first paragraph entitled “GRFT/CG FDIs protect mice against HSV-2 G and HPV16 PsV” (line 139-141).

In conclusion, the data have merit, but citation of supporting unpublished data is unacceptable and the manuscript does not stand on its own without them. I suggest the supporting information should be provided in a Supplementary Information section. I also suggest that the manuscript would benefit from careful proofreading.

We believe that we have sufficiently addressed Dr. Palmer’s concerns through (1) the removal of all reference to unpublished data, (2) the inclusion of these data within the Supplementary Materials section, and (3) careful proofreading of the manuscript.

Reviewer 2

In this paper, the authors have undertaken a careful evaluation of a griffithsin (GRFT)/carrageenan (CG) fast dissolving insert (FDI) for the prevention of SHIV, HSV-2 and HPV infections. Efficacy studies were undertaken in a non-human primate model (for SHIV efficacy) and the mouse model (for HSV-2 and HPV infection). The studies were well designed, appropriate, and conclusive and demonstrate significant efficacy for these three pathogens. The development of a multi-purpose FDI is highly responsive to the current direction of HIV prevention research. Additional data showing persistence of the antiviral efficacy out to 8 hours post dosing is also very encouraging for a pericoital product.

I have no concerns about the technical aspects of the study but do think that the paper would be strengthened by a brief discussion of what (if anything) is known about the safety and PK profile of repeated exposure to griffithsin/carrageenan FDI or gel. The authors mention that a Phase 1 gel study is ongoing and so I assume that there are repeat dosing GLP animal toxicology data. One potential concern related to the use of protein based products is the possibility that local or systemic absorption of the product might induce immunological responses that could reduce product efficacy. The authors state that single dose administration was not associated with systemic PK exposure but it would be interesting to know if there are any repeat dose data.

We appreciate the Reviewer's interest in these data and have decided to include the data in the manuscript. (See also the response to Reviewer 1). We have included data from repeated exposure safety/toxicology/PK studies using intravenous and vaginal delivery of GRFT and GRFT/CG gel in mice, rats, and rabbits as Supplementary Table 3 and associated brief text at the end of the Results in a new section entitled "*GRFT remains safe and minimally absorbed after repeated application*" (line 156-167). These studies demonstrate overall (1) little to no safety signals in the form of vaginal irritation or systemic toxicity even after intravenous delivery, (2) little to no systemic accumulation of GRFT after vaginal application, (3) no detection of anti-drug antibodies following intravenous exposure.

The authors mention that the product was stored at 2-8°C until the FDIs were used in the challenge experiments. Does this imply that the product would require commercial storage in this temperature range.

We stored the FDIs at 2-8C because at the time of planning and initiating the macaque studies, we wanted to minimize possible degradation prior to the start of the studies but we did not yet have extended stability data at room temperature. We now have these data, and FDIs for commercial use would be stored at ambient temperature. Since these data are published in Lal, et al. 2018 J. Pharm Sci., we have added reference to this in the manuscript in the first paragraph of the Methods section ("*Fast dissolving inserts*") on line 276-278.

In addition, there are reports that (1) GRFT is prone to oxidation, by both hydrogen peroxide exposure and human cervicovaginal secretion exposure; (2) Methionine at position 78 in the amino acid sequence of GRFT is oxidized. The authors do not comment on the stability of the FDI to oxidation and it would be useful to have this information if it is available for the GRFT/CG FDI.

The Reviewer is clearly quite familiar with the challenges of developing GRFT as a microbicide and rightly mentions this important issue. Since the information is indeed available within the formulation manuscript Lal, et al. 2018 J. Pharm Sci., we have added a comment about the stability of the formulation in the Introduction (last paragraph –line 84-85) and Discussion (first paragraph –line 178-182).

Reviewer 3

In the manuscript "Griffithsin/Carrageenan Fast Dissolving Inserts Prevent SHIV, HSV-2 and HPV Infections in vivo", Derby et al present a series of experiments showing the efficacy of a new antimicrobial compound/formulation in preventing several vaginal viral infections. Here they report the effects of a new Griffithsin/Carrageenan fast dissolving insert in macaque and mouse models of SHIV, HSV-2, and HPV infections. Both compounds have been previously reported to provide some protective efficacy in vitro and in vivo against the three viruses tested here. Interestingly, the GR/CG FDIs provide drug intravaginally for 8 hours (and in some cases up to 24 hours) after insertion without altering vaginal pH and without entering circulation. The authors first present the results of an animal study where 8 out of 10 RMs were protected from a single moderate dose vaginal SHIVsf162P3 challenge. The authors then describe two mouse studies showing protection from high dose (lethal) HSV-2 challenge (63%) and HPV16 pseudovirus infection (100%).

This is a highly important, well-written study of a new antimicrobial treatment that could provide women across the globe with an effective, discreet means of protecting themselves from HIV transmission. Acknowledging space limitations, this manuscript could be improved by addressing the following comments.

Minor Comments:

1. 647 TCID50 isn't high dose viral challenge. This is a moderate dose at best. If you are comparing the dose to what you expect semen viral titers to be then make this clear and cite your sources.

The point is well taken, and we have modified the text throughout to remove reference to the inoculum as a "high dose" (see removal compared to original version on line 85). Instead of focusing on the dose of virus being high, per se, we have added a line stating that the inoculum was chosen to result in 100% infection frequency in controls after a single challenge and thus it is a highly susceptible model that represents an enhanced susceptibility compared with human exposure. See line 185-188. We have also cited sources as requested for the inocula detected in semen. See second paragraph of Discussion on line 187.

2. It isn't clear from the methods section how many times this virus has been passaged in vitro since its original growth. Mutations will occur in the culture. Thus the extent to which these mutations will have any impact on transmission is actually up in the air (refer to page 8 lines 10-11.) It shouldn't be hard to sequence envelope in your virus stock and determine whether the virus lost glycans over its passage history.

The Reviewer raises an important point. In answer to the question about passage history, the virus stock used to challenge animals herein was a third-generation passage off of virus obtained from the NIH Division of AIDS. We have added relevant information in the Methods ("In vivo SHIV SF162P3 Challenge" section on line 301-302 and 306-307).

We have sequenced the env of the stock virus through a clonal sequencing approach and found that there is loss of only one potential N-linked glycosylation site (PNG) in our stock compared with published env sequence of SHIV SF162P3 (N230 based on HxB2 numbering, position 228 in SHIV SF162P3). The sequence data (10 clones of stock virus) are now provided in Supplementary Figure 1. Mutation of the PNG at this position was associated with increased infectivity of cell free virus and modestly increased cell-cell transmission (Mathys, 2015). This may have helped in the high infection rate we observed in vivo with this stock. Loss of N230 in combination with other PNGs has also been associated with development of resistance to lectin-mediated HIV inhibition in four clade C isolates (Alexandre, 2013, Virology).

Importantly, if loss of the glycan had any effect on GRFT activity in our study, it would have been to decrease activity both through increasing virus infectivity and decreasing GRFT efficacy. Yet we still saw highly significant 80% inhibition of infection, further supporting the excellent activity of GRFT in vivo.

Notably, we were only able to obtain a few plasma virus sequences from infected macaques (5 clones from 1 of 2 GRFT/CG exposed animals; 1 clone from 1 of 10 CG exposed animals) and for this reason mentioned the data as preliminary and “data not shown” in the originally submitted version. Nonetheless, we have decided to include these data now within the Figure to highlight the lack of GRFT-mediated selection.

Discussion of the above points is now included in the Discussion section, paragraph 4 on line 219-233.

3. HPV pseudovirus didn't infect 100% of controls suggesting that the virus stock had a low titer, so is sterilization actually 100%?

For clarity, we have removed reference to “complete/100% protection” to describe the HPV protection throughout the manuscript and instead describe the protection as “significant”, which indeed it is by ANOVA testing. Wording is corrected in the Abstract (line 39-40) and corrected and clarified in the Results in the third paragraph of the section “GRFT/CG FDIs protect mice against HSV-2 G and HPV16 PsV” (line 148-150). In the Results, we added the specific number of animals infected in the control vs. GRFT/CG groups for added clarity (line 148-150).

4. The effectiveness of any treatment focused on entry will depend largely on its ability to neutralize multiple subtypes and multiple different envelope sequences within subtypes. Is there evidence that some envelopes may be less susceptible? Please discuss this in the manuscript.

The Reviewer raises an excellent point – for entry blockers to make an impact on the HIV epidemic, they must be able to act on diverse envelopes. GRFT is well positioned because published data (O’Keefe, PNAS, 2009) show that GRFT is highly potent against transmitted isolates from clades B and C (4 isolates tested per clade; $EC_{50} < 3\text{ng/ml}$ for 4 of 4 clade B and 3 of 4 clade C; the fourth clade C had $EC_{50} = 10\text{ng/ml}$). GRFT also exhibited activity against transmitted isolates from clade A though activity was overall somewhat less, linked with fewer external facing N-linked glycosylation sites on the clade A envelopes tested ($EC_{50} < 3\text{ng/ml}$ for 1 of 4 clade A and 70-150ng/ml for 3 of 4). GRFT also hit more isolates than did any of the lead bNAbs from that decade (IgG1b12, 2F5, 4E10, 2G12).

Due to the overwhelming prevalence of clade C in southern Africa, the region of the world unarguably hardest hit by the HIV epidemic, we believe that GRFT can make an important contribution to preventing transmission. In addition, published data show that the development of resistance to GRFT in clade C isolates is slow (Alexandre, 2013). We have added a paragraph to the Discussion – new paragraph 6 on line 242-252 – on this topic.

Grammatical comments:

1. Methods incomplete: cytokine assays, menstrual cycle determination, expand (briefly) pseudovirus assay / detection.

Please see below for responses to each of the comments

“...cytokine assays...”

The description of methods for the cytokine assay (performed by Luminex) is within the section of Methods on Safety and PK within the overarching section entitled “Anti-SHIV activity in macaques”. However, as the data are shown only in the Supplementary Materials, we have moved the cytokine data (along with the vaginal pH data) to the Supplementary Methods section on line 577-585. If the Reviewer still finds this section incomplete, we are happy to provide more information.

“...menstrual cycle determination...”

We did not determine the menstrual cycles of the macaques in the study. Reference in the Discussion to the menstrual cycles of infected and uninfected macaques in the challenge study being similar (line 211) was based on the fact that all animals were treated with DMPA and thus were hormonally synchronized.

“...expand (briefly) pseudovirus assay/detection.”

We realize that the HPV16 PsV infection assay in mice is not widely used and should be described in somewhat more detail. In addition, when referencing our prior publications on this assay, we neglected to also include citation of the original paper describing the approach (Roberts, et al 2007). We have now included the Roberts citation (now reference 22) and also expanded the description in the relevant section of the Methods (line 355-359).

2. Define acronyms at first reference: DMPA, CVL

We have introduced the acronym definitions as appropriate: DMPA (line 97), CVL (line 114), and we also found we missed the acronym for EC50 (line 69)

3. page 9: "Anti-SHIV activity in macaques" in italics doesn't have a paragraph associated with it.

We initially divided the Methods into 5 major sections:

- 1- Fast dissolving inserts
- 2- Anti-SHIV activity in macaques
- 3- Anti-HSV-2 and HPV16 PsV activity in mice
- 4- GRFT ELISA
- 5- Statistics

The second and third sections were further subdivided:

- 2- Anti-SHIV activity in macaques
 - a- Ethics and animal care
 - b- In vivo SHIV SF162P3 challenge
 - c- Safety and PK
 - d- In vitro antiviral activity of in vivo-delivered GRFT
- 3- Anti-HSV-2 and HPV16 PsV activity in mice
 - a- Ethics and animal care
 - b- HSV-2 challenge
 - c- HPV16 PsV challenge
 - d- GRFT PK

Of note, we have retitled 2c to be “GRFT PK” (line 321) since we moved the safety data into Supplementary Materials.

Reviewers' Comments:

Reviewer #1:

Remarks to the Author:

The authors have submitted a revised manuscript that properly addresses the reviewers' comments. The data presentation is good, and supportive of the conclusions of the study. I congratulate the authors and recommend that the manuscript is suitable for publication in Nature Communications.

Kenneth Palmer

Reviewer #2:

Remarks to the Author:

I think that the authors have responded adequately to the reviewers' comments. The article is much improved and I think that that it would be reasonable to accept this article for publication in your journal.

Reviewer #3:

Remarks to the Author:

The authors have sufficiently addressed my critiques from the previous round of review. This manuscript should be accepted for publication.

Thomas H. Vanderford